# Optimal mass variables for semivisible jets

**Kevin Pedro⋆ and Prasanth Shyamsundar**

Fermi National Accelerator Laboratory, Batavia, IL 60510, USA

⋆ pedrok@fnal.gov

## Abstract

Strongly coupled hidden sector theories predict collider production of invisible, composite dark matter candidates mixed with standard model hadrons in the form of semivisible jets. Classical mass reconstruction techniques may not be optimal for these unusual topologies, in which the missing transverse momentum comes from massive particles and has a nontrivial relationship to the visible jet momentum. We apply the artificial event variable network, a semisupervised, interpretable machine learning technique that uses an information bottleneck, to derive superior mass reconstruction functions for several cases of resonant semivisible jet production. We demonstrate that the technique can extrapolate to unknown signal model parameter values. We further demonstrate the viability of conducting an actual search for new physics using this method, by applying the learned functions to standard model background events from quantum chromodynamics.

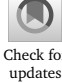
# 1  Introduction

Dark matter (DM) is one of the clearest indicators of the existence of physics beyond the standard model (SM). Its gravitational interactions have been observed in numerous astrophysical contexts, spanning rotation curves [1, 2], lensing [3], galaxy cluster collisions [4], and the cosmic microwave background [5]. However, attempts to detect dark matter through weak interactions with SM particles, whether via direct interactions, annihilation, or collider production, have so far been unsuccessful.

Dark matter may still be accessible at current-generation experiments if its nature is different from the simplest models of weakly interacting massive particles (WIMPs). In this paper, we consider the possibility that dark matter is primarily composed of composite particles, much like visible matter. Composite dark matter may arise from a hidden valley (HV) that communicates with the SM only via weakly-interacting heavy mediator particles [6]. In particular, this hidden valley may be a strongly coupled hidden sector with multiple species of dark quarks $\chi$ charged under a new, confining dark force carried by dark gluons, which form dark hadrons. The new dark force can be described as "dark QCD", in analogy with SM quantum chromodynamics. Some of the dark hadrons are stable and act as dark matter candidates, while the unstable varieties decay to SM particles such as quark-antiquark pairs.

Such models are more discoverable at colliders than direct detection or annihilation experiments. Direct detection of dark hadrons is suppressed below the neutrino floor [7], and annihilation is expected to be rare for any form of dark matter arising from an asymmetry, whether or not it is composite [8]. Assuming collider production of the hidden valley mediator and prompt decays of the unstable dark hadrons, the final state may include "semivisible" jets containing both visible SM and invisible DM particles [9]. It has been shown that the observed dark matter relic density can be obtained from this class of models [10, 11]. Other possible signatures, not considered here, include emerging jets, when the unstable dark hadrons are long-lived [12], or soft unclustered energy patterns, when the 't Hooft coupling is large and wide-angle radiation is not suppressed [13].

The first experimental search for semivisible jets [14] by the CMS experiment at the LHC considered $s$-channel production with a heavy leptophobic Z′ boson mediator, shown in Fig. 1 (left). This search placed limits on the product of the Z′ production cross section and the branching fraction to dark quarks Z′ → $\chi\overline{\chi}$, which can be translated into limits on $m_{Z'}$ for specific values of the Z′ couplings to SM quarks ($g_q$) and dark quarks ($g_\chi$). However, the benchmark values for these couplings, and therefore the predicted cross sections and branching fractions, are ultimately arbitrary. Hence, a Z′, or another $s$-channel mediator, accessible at the LHC may still exist, just interacting too weakly to be produced in detectable quantities in the available datasets. This search used the transverse mass $M_T$ of the dijet system and the missing transverse momentum $\not{p}_T$ to reconstruct the Z′. In simpler topologies, with massless invisible particles well-separated from massive visible particles, $M_T$ is known to be optimal,

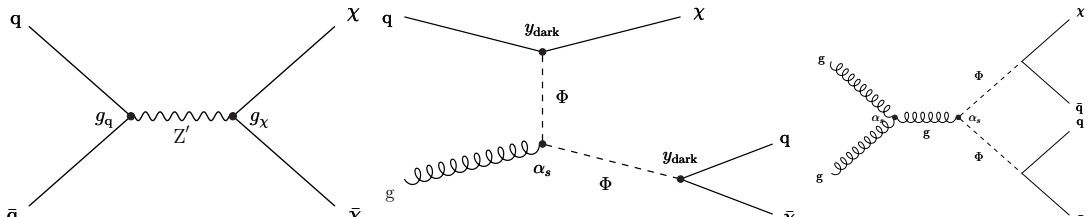

Figure 1: Representative Feynman diagrams for leading-order production of a $Z'$ boson decaying to dark quarks $\chi$ (left), a single $\Phi$ boson associated with a dark quark and decaying to an SM quark and a dark quark (center), and a pair of $\Phi$ bosons each decaying to a dark quark and an SM quark (right).

but it is an open question if it remains optimal in more complicated topologies, such as the semivisible jet case described here.

In addition to $Z'$ bosons and other $s$-channel mediators, another notable possibility is a bifundamental scalar particle $\Phi$, charged under both SM QCD and dark QCD [7, 15]. $\Phi$ has Yukawa couplings $y_{\text{dark}}$ between dark quarks and SM quarks, leading to different final states than the $Z'$ $s$-channel case. Relevant to the goal of this paper, $\Phi$ can be produced singly or in pairs, as shown in Fig. 1 (center, right). The optimal mass reconstruction for such final states has not been thoroughly investigated. The first experimental search for emerging jets [16], also from CMS, considers the pair production of $\Phi$, but does not attempt to reconstruct its mass. A recent search from the ATLAS experiment sets limits on certain models of non-resonant $t$-channel SVJ production via $\Phi$ [17], but does not target resonant production. For other final states with pairs of massive particles decaying to both visible and invisible particles, the $M_{\text{T2}}$ variable has been shown to be useful [18], though not necessarily optimal or unique [19].

Here, we employ a new machine learning technique to derive optimal mass reconstructions for these three final states: $Z'$, single $\Phi$, and pairs of $\Phi$. This technique, called the event variable network (EVN), was first introduced in Ref. [20]. It uses an information bottleneck to learn a generalized function whose output, the artificial event variable (AEV or simply $\vec{V}$), maximizes the mutual information with a target parameter. Because it is only trained on signal models, with no information provided about background processes, it is semisupervised; and because the target parameter is a physically meaningful quantity, the output of the network is similarly physically meaningful, leading to an interpretable result. In particular, Ref. [20] shows that for a fully visible final state where the target parameter is the theoretical mediator mass, the learned function is equivalent to the invariant mass calculation, and therefore we expect the EVN to produce optimal mass estimators in semivisible final states, as well. We apply the EVN to each final state and compare the resulting AEV with existing classical, analytical mass reconstruction algorithms. We also demonstrate the generalization properties of the EVN by testing it on signal models with different parameter values not used during the network training, as well as on simulations of SM QCD, the primary background for semivisible jet signals.

## 2 Models and simulation

The dark sector signal model is implemented following the CMS search [14], which was originally based on the model in Ref. [9]. The important parameter values and ranges are summarized here, while more details can be found in the aforementioned references. The number of dark colors is $N_c^{\text{dark}} = 2$ and the number of dark quark flavors is $N_f^{\text{dark}} = 2$. The dark hadron mass is chosen to be $m_{\text{dark}} = 20\,\text{GeV}$, and the dark quark mass is set to $m_\chi = m_{\text{dark}}/2$. The cou-

pling scale of the dark force is defined in terms of the dark hadron mass, $\Lambda_{\text{dark}} = 3.2(m_{\text{dark}})^{0.8}$, which is an empirical relation that approximately maximizes the dark hadron multiplicity in the dark shower. The dark hadron mass scale does not significantly impact the event-level kinematic quantities that are used in this paper, so we do not consider variations of it or the parameters with related values. The invisible fraction $r_{\text{inv}}$, defined as the fraction of dark hadrons that are stable, is the most novel and impactful parameter of the signal model, and can take any value between 0 and 1. Any unstable dark hadrons can decay to pairs of any available species of SM quarks, where availability is defined by the condition $m_{\text{dark}} \geq 2m_{\text{q}}$. The unstable vector dark hadrons decay democratically, while the unstable pseudoscalar dark hadrons decay via a mass insertion, therefore preferring the most massive available SM quarks. The probability of producing a vector dark hadron, as opposed to a pseudoscalar, is set to 0.75. For Z$'$ production, the couplings are set to $g_{\text{q}} = 0.25$ and $g_{\chi} = 1.0/\sqrt{N_c^{\text{dark}}N_f^{\text{dark}}} = 0.5$, which implies $\mathcal{B}_{\text{dark}} = 47\%$, consistent with the LHC DM Working Group benchmark [21]. For $\Phi$ production, the Yukawa couplings are set to $y_{\text{dark}} = 1.0$ for all species of SM quarks and dark quarks.

The Z$'$ signal events are generated using PYTHIA version 8.230 [22], with the dedicated HV module used for showering and hadronization in the dark sector. The HV module simulates the dark sector dynamics using the Lund string model [23,24]; we use the default settings for the empirical parameters in this model, which are the values tuned for SM QCD. Different parameter values, or even a different dynamical model or generator software, could change the dark sector dynamics. However, such variations generally impact the formation and substructure of jets, rather than their final four-momenta, and therefore are not expected to have a significant impact on the event-level mass reconstruction pursued here. The NNPDF3.1 leading order (LO) parton density function (PDF) [25] and the CP2 underlying event tune [26] are used. The events are generated with $m_{Z'}$ values ranging from 500 to 5000 GeV in steps of 100 GeV, and $r_{\text{inv}}$ values of 0.1, 0.3, 0.5, and 0.7; $m_{\text{dark}}$ is set to 20 GeV. Approximately 12000 events per signal model are generated for the models with $r_{\text{inv}} = 0.3$, while 6000 events per signal model are generated for the models with other $r_{\text{inv}}$ values. The Z$'$ cross section is computed at next-to-leading-order (NLO).

The $\Phi$ signal events are generated using MADGRAPH5_aMC@NLO version 2.6.5 [27] at LO, with the new particles and couplings implemented via FEYNRULES [28] following Ref. [7]. Showering and hadronization are performed using PYTHIA version 8.240, and the MLM matching procedure is employed to eliminate double counting of radiation [29]. The NNPDF3.1 next-to-next-to-leading-order (NNLO) positive-definite PDF and the CP5 tune are used. The absolute and relative cross sections for processes involving $\Phi$, split by the number of resonant mediators $n_{\Phi}$, are taken from MADGRAPH5_aMC@NLO and shown in Fig. 2 for the chosen Yukawa coupling value $y_{\text{dark}} = 1.0$. Nonresonant production depends more strongly on $y_{\text{dark}}$ than single production, while pair production primarily depends only on $\alpha_{\text{S}}$; therefore, the relative fractions of the resonant production modes would increase for smaller $y_{\text{dark}}$ values and decrease for larger $y_{\text{dark}}$ values. The events with $n_{\Phi} = 1$ and $n_{\Phi} = 2$ are generated with $m_{\Phi}$ values from 500 to 2000 GeV in steps of 100 GeV; $r_{\text{inv}}$ is set to 0.3 and $m_{\text{dark}}$ is set to 20 GeV. Approximately 6000 events per signal model are generated.

The QCD multijet background sample is generated with PYTHIA version 8.205, using a biased sampling to generate a roughly flat distribution in $15 < \hat{p}_{\text{T}} < 7000$ GeV. A weight is subsequently applied to produce a physical distribution by inverting the sampling bias. The NNPDF2.3 LO PDF [30] and the CUETP8M1 tune [31] are used. Approximately 10 million QCD events are generated in the flattened $\hat{p}_{\text{T}}$ space.

Given the goal of this paper to learn about fundamental kinematic relationships, no detector simulation is performed and generator-level quantities are used. Visible particles denoted as stable by the generator are clustered into jets J using the anti-$k_{\text{T}}$ algorithm [32], imple-

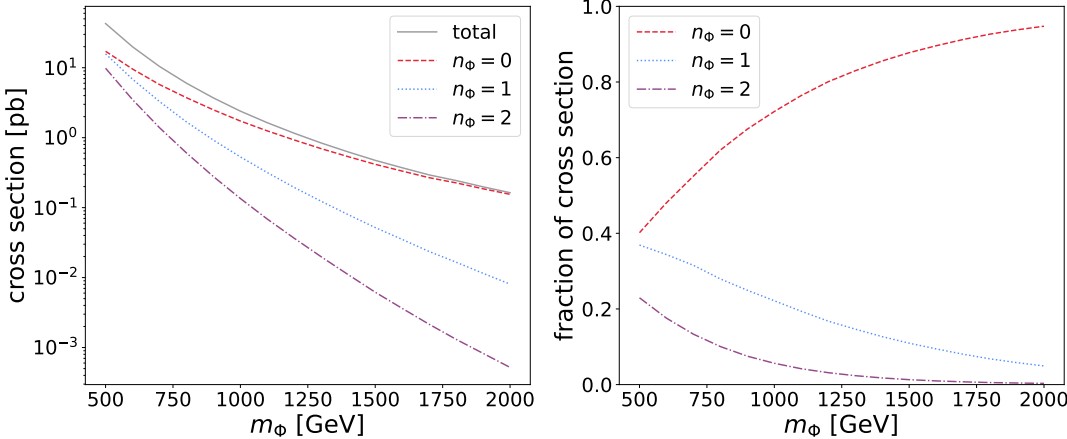

Figure 2: The absolute (left) and relative (right) cross sections for processes involving $\Phi$ mediators.

mented in the FASTJET software [33]. A distance parameter of $R = 0.8$ is used because the additional decay step from the dark sector to the SM leads semivisible jets to have a broader spread in their visible constituents than SM jets. The jets are sorted by their transverse momentum. The missing transverse momentum is computed as the negative of the vector sum of the transverse momentum vectors of all visible stable particles. Computations related to the $M_{\mathrm{T2}}$ variable are performed using Ref. [34].

## 3 Architecture and training

The EVN is trained using a composite neural network; its inputs, structure, and outputs are shown in Fig. 3. The event data are prepared for training the network by splitting the simulated events into two classes. In class 1, the values of the theory parameters $\vec{\theta}$ correspond to the kinematic input variables $\vec{x}$, while in class 0, the values of $\vec{\theta}$ are random and do not correspond to $\vec{x}$. The EVN is the first component and serves as the information bottleneck. It is a fully-connected network that takes the inputs $\vec{x}$ and produces output $\vec{V}$, which is typically, though not necessarily, a single value per event. The second component is a classifier, also a fully-

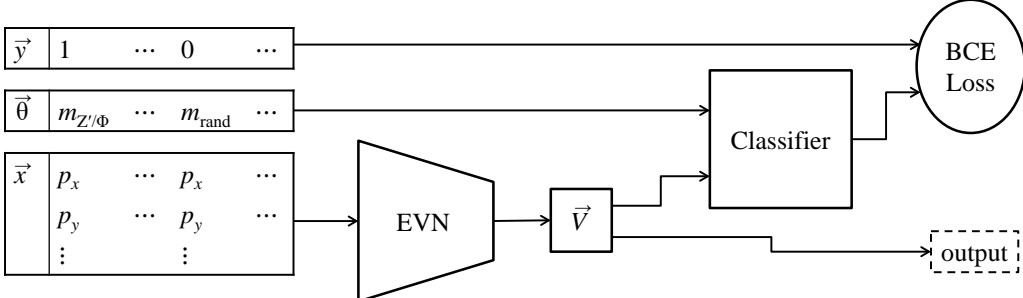

Figure 3: A diagram of the composite network architecture, showing the inputs, neural network blocks, output, and loss function. Example inputs are indicated for the representative case of reconstructing a mass value using four-vector values; $m_{\mathrm{rand}}$ represents the random theory parameter values assigned for class 0 events, as described in the text.

Table 1: The hyperparameter values used to train the composite network.

| Parameter | Value |
| --- | --- |
| % events used for training | 64 |
| % events used for validation | 16 |
| % events used for testing | 20 |
| Batch size | 5000 |
| Epochs | 100 |
| EVN layer sizes | 128, 64, 64, 64, 32 |
| Classifier layer sizes | 16, 16, 16 |
| Learning rate | 0.001 |

connected network, which combines the EVN output $\vec{V}$ and $\vec{\theta}$ to determine the event class, with the binary crossentropy (BCE) loss as its objective to minimize. Because the two components of the network are jointly optimized, the EVN learns an optimal and general function to combine the inputs into $\vec{V}$; this distinguishes the technique from a simple regression. In class 0 events, $\vec{\theta}$ and $\vec{x}$ are independent, so their joint distribution is simply the factorized distribution $p_{\vec{V}} p_{\vec{\theta}}$, the product of their independent probabilities. In class 1 events, $\vec{x}$ arises from $\vec{\theta}$, so their joint distribution $p_{\vec{V}, \vec{\theta}}$ depends on their conditional distribution. The classification process maximizes the ability to distinguish between the two classes, therefore maximizing the mutual information between $\vec{\theta}$ and $\vec{V}$, defined as $I(\vec{V}; \vec{\theta}) = \int_{\vec{V}} \int_{\vec{\theta}} p_{\vec{V}, \vec{\theta}} \log[p_{\vec{V}, \vec{\theta}}/(p_{\vec{V}} p_{\vec{\theta}})]$. A more thorough derivation of this network is given in Ref. [20].

The hyperparameters used to train the composite network are summarized in Table 1. The datasets are split into training, validation, and testing portions; the validation portion is used to check for overtraining by comparing the loss values, while the testing dataset is used for calibration and statistical assessments, as described in Section 4. In addition, a second, independent testing dataset is employed for more in-depth physical comparisons. While an exhaustive hyperparameter scan is not performed, we note a few findings. The chosen values for batch and layer sizes perform better than smaller values, while larger values run the risk of memorization or overtraining. The total number of trainable parameters in the network is 20706. The chosen learning rate performs better than larger values. Extending the number of epochs further does not improve the result. The Adam optimizer [35] is used, and ReLU activation is applied to all internal layers of both fully-connected networks. The training takes a few minutes on an Nvidia RTX 2080 Super, which is a typical consumer GPU.

We also note one difference with respect to Ref. [20]: in class 1 events, as described in Section 2, mediator mass values are generated in discrete steps of 100 GeV rather than continuously. (The "fake" mass values assigned to class 0 events are still generated continuously.) This performs as well as the continuous approach as long as balance is maintained between the number of events with each discrete value in the overall dataset.

# 4 $Z'$ production

## 4.1 Mass variables

As discussed previously, the variable traditionally used to reconstruct the mediator mass for $Z' \to \chi \overline{\chi}$ is the transverse mass. This variable is defined as

$$M_{\mathrm{T}}^2 = \left(E_{\mathrm{T,JJ}} + \not{E}_{\mathrm{T}}\right)^2 - \left(\vec{p}_{\mathrm{T,JJ}} + \vec{\not{p}}_{\mathrm{T}}\right)^2 = m_{\mathrm{JJ}}^2 + 2\left(\sqrt{m_{\mathrm{JJ}}^2 + p_{\mathrm{T,JJ}}^2}\,\not{p}_{\mathrm{T}} - \vec{p}_{\mathrm{T,JJ}} \cdot \vec{\not{p}}_{\mathrm{T}}\right),$$

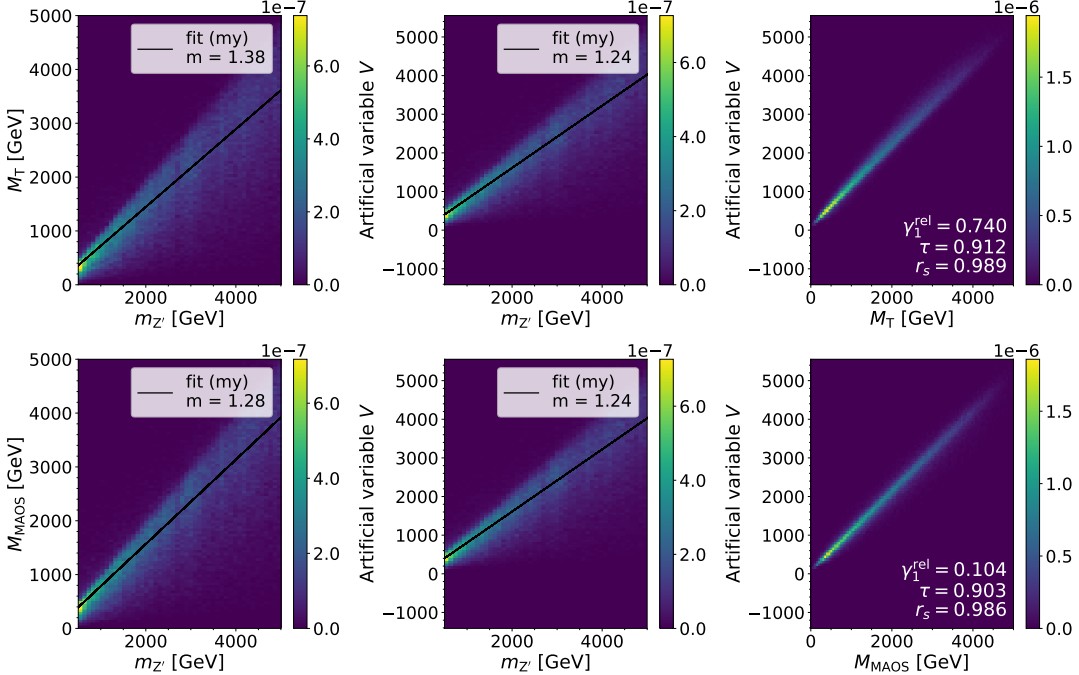

Figure 4: Top: Correlations between $M_{\rm T}$, $V$, and $m_{\rm Z'}$. Bottom: Correlations between $M_{\rm MAOS}$, $V$, and $m_{\rm Z'}$. The black lines show the calibrations of the reconstructed variables to the scale of $m_{\rm Z'}$, where $m$ is the slope of the fit $x = my$.

where $E_{\rm T,JJ} = \sqrt{m_{\rm JJ}^2 + p_{\rm T,JJ}^2}$ is the energy of the massive dijet system, with $m_{\rm JJ}$ and $p_{\rm T,JJ}$ the invariant mass and transverse momentum of that system, respectively, and $\displaystyle{\not}E_{\rm T} = \displaystyle{\not}p_{\rm T}$ is the energy of the invisible system, which is assumed to be massless. However, there is actually another classical variable that can improve on $M_{\rm T}$, using the $M_{\rm T2}$-Assisted On Shell (MAOS) technique [36]. MAOS was originally derived for cases like H $\rightarrow$ WW, where one resonance decays to two resonances, each of which decays to a visible and invisible component. The $M_{\rm T2}$ algorithm divides the missing transverse momentum into two parts, one corresponding to each visible component of the event, $p^a$ and $p^b$. MAOS promotes these two $\vec{\displaystyle{\not}p}_{\rm T}$ two-vectors to four-vectors by assigning $\displaystyle{\not}m_i = 0$ and $\displaystyle{\not}p_z^i = p_z^i(\displaystyle{\not}p_{\rm T}^i/p_{\rm T}^i)$, where $i = a, b$, and then computes the full invariant mass of the visible and invisible components, defined as $M_{\rm MAOS}$.

As a first test, we apply the EVN to Z$' \rightarrow \chi\bar{\chi}$ events with $m_{\rm dark} = 20\,{\rm GeV}$ and $r_{\rm inv} = 0.3$. The network inputs $\vec{x}$ include the leading two jet four-vectors, which are represented as $(E, p_x, p_y, p_z)$ for each jet, and the $\vec{\displaystyle{\not}p}_{\rm T}$ two-vector, which is represented as $(\displaystyle{\not}p_x, \displaystyle{\not}p_y)$. The correlations between the network output $V$, the two classical variables $M_{\rm T}$ and $M_{\rm MAOS}$, and the theory parameter $m_{\rm Z'}$ are shown in Fig. 4. These plots additionally display the calibration procedure, which will be used in subsequent plots. Calibration is necessary to make direct comparisons between the classical variables and $V$, because the absolute scale of $V$ is not fixed. For consistency, all reconstructed variables are calibrated the same way, via a linear fit to the theory parameter. (The constant term in the linear fit is found to be very small and is therefore neglected.) The correlation plots, and therefore the calibration procedure, use the independent dataset composed of the 20% of events reserved for testing.

We compare the classical and learned variables using several statistics, including Kendall's $\tau$ and Spearman's rank coefficient $r_s$. $V$ is highly correlated with, but not identical to, both $M_{\rm T}$ and $M_{\rm MAOS}$. Additionally, we introduce a new statistic called the relative skewness of the

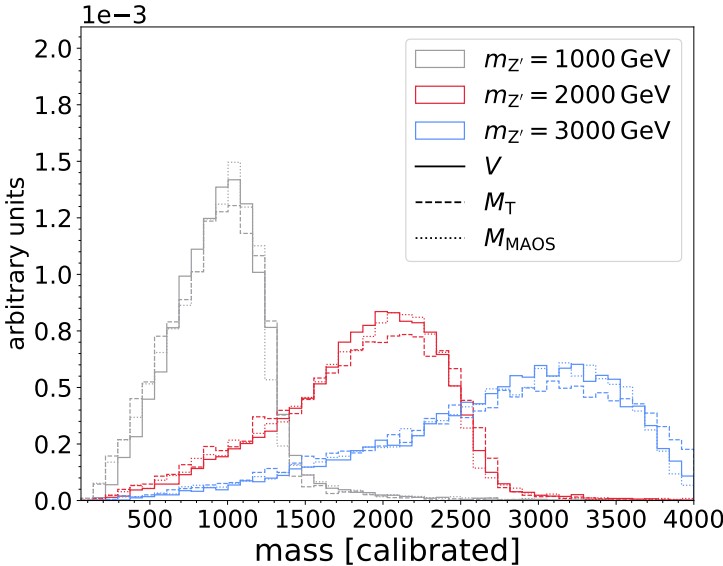

Figure 5: The distributions of $M_T$, $M_{MAOS}$, and $V$ for several $m_{Z'}$ values, normalized to unit area.

correlation or $\gamma_1^{rel}$, which is defined as follows:

1. perform a linear fit $Y = \alpha X$, where $X$ and $Y$ are two datasets, resulting in the best-fit value $\hat{\alpha}$ for $\alpha$;
2. define $\tilde{Y} = Y/\hat{\alpha}$, so that the linear fit relationship becomes $\tilde{Y} = X + $ residual;
3. define the line $\tilde{Y} = X$ as the new independent axis, corresponding to a 45° rotation into new coordinates $X' = \frac{1}{\sqrt{2}}(X + \tilde{Y})$, $Y' = \frac{1}{\sqrt{2}}(\tilde{Y} - X)$;
4. compute $\gamma_1^{rel}$ as the Fisher-Pearson skewness coefficient of $Y'/X'$, the relative displacement from the $X'$ axis.

The utility of the new statistic can be observed by comparing the values of the other correlation statistics for $V$ vs. $M_T$ and $V$ vs. $M_{MAOS}$, which are very similar. In contrast, $\gamma_1^{rel}$ is much smaller for $M_{MAOS}$ than for $M_T$. This indicates that $V$ tends to be larger than $M_T$, but there is little bias in the difference between $V$ and $M_{MAOS}$.

The similarity between $V$ and $M_{MAOS}$ can be further observed in Fig. 5, which compares distributions of each variable for several $m_{Z'}$ values. The second independent testing dataset is used in this figure. As noted above, the calibrated version of each variable is used to avoid spurious differences in scale. It can be seen that the distribution of $V$ is similar in shape to $M_{MAOS}$ and is indeed narrower than $M_T$. Therefore, we conclude that the $M_{MAOS}$ procedure results in a nearly optimal reconstruction under the given assumptions. It is expected that $V$ and $M_{MAOS}$ would not be identical, because $M_{MAOS}$ depends on the result of the $M_{T2}$ calculation, which is not a singularity variable [19].

## 4.2 Discovery potential

Subsequently, we demonstrate the utility of the learned variable $V$ for discovery of this channel of semivisible jet production. We compare the sensitivity, defined as $Q = S/\sqrt{B}$ where $S$ and $B$ are the signal and background yields, from binned mass distributions. Examining the strategy from Ref. [14], rejection of the major background from QCD multijets is primarily accomplished via the "transverse ratio", a relative variable defined by $R_T = \not{p}_T/M$, where $M$ is the reconstructed mass variable. Here, we use generator-level simulations of QCD, which lack instrumental effects such as dead calorimeter cells that induce artificial $\not{p}_T$. Therefore,

we apply a selection $R_T > 0.1$, which is looser than the experimental thresholds of 0.15–0.25. The lower $\not{p}_T$ values in the generator-level background imply that fewer events have high $R_T$ values, so this looser selection provides a level of background rejection similar to the realistic analysis. We also require $p_T > 200\,\mathrm{GeV}$ for the leading two jets; all other selections are omitted for simplicity. For each reconstructed mass variable, the selection is applied using $R_T$ defined in terms of that variable. The resulting distributions are shown in Fig. 6, with both the signal and background samples normalized to an integrated luminosity of $138\,\mathrm{fb}^{-1}$. The background distributions are very similar for all three reconstructed mass variables; we emphasize here that the EVN was trained only on signal events and has no explicit knowledge of QCD multijet background events. The bin-by-bin significance ratios can also be examined, noting that the bin at the peak of the distribution provides the largest contribution to the overall sensitivity. It can be seen that the sensitivity of the artificial variable in this peak bin is approximately 30% higher than $M_T$ and similar to $M_{\mathrm{MAOS}}$ for the higher $m_{Z'}$ values. The overall significance for each signal model and mass variable can be approximated by adding the significance of each bin in quadrature. This calculation shows that the artificial variable has a 3–5% higher significance than $M_T$, while $M_{\mathrm{MAOS}}$ has only a $\approx 1\%$ higher significance. Because the improvements are minor, searches may still prefer to use the simpler $M_T$ variable, but in the event of a discovery, the $Z'$ mass could be measured with better resolution using the artificial variable.

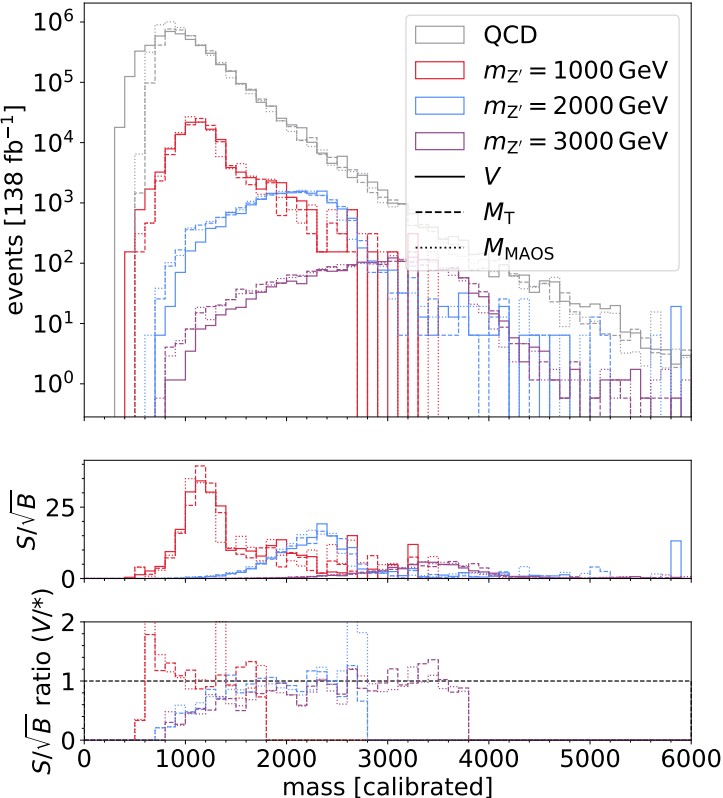

Figure 6: The distributions of each reconstructed mass variable for $Z'$ and QCD multijet processes. The middle pane shows the significance, and the bottom pane shows the ratio of the significance, comparing the two classical variables to $V$. The distributions in the bottom pane are truncated to eliminate statistical fluctuations from limited numbers of events in the high tails of the signal mass distributions.

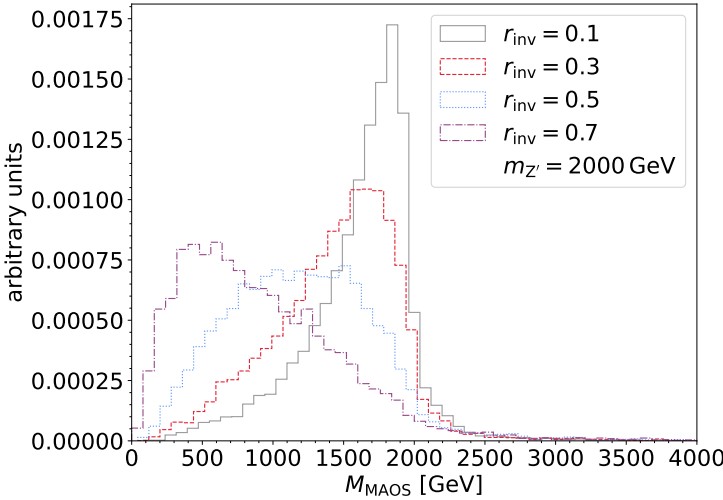

Figure 7: The distributions of $M_{\mathrm{MAOS}}$ for different $r_{\mathrm{inv}}$ values, using samples with $m_{Z'} = 2000\,\mathrm{GeV}$.

## 4.3 Varying $r_{\mathrm{inv}}$

The $r_{\mathrm{inv}}$ parameter has a large effect on the potential accuracy of the $Z'$ mass reconstruction, as shown in Fig. 7 for $M_{\mathrm{MAOS}}$. To assess the generalization properties of the EVN, we train four separate instances of the network, corresponding to distinct signal samples with $r_{\mathrm{inv}}$ values of 0.1, 0.3, 0.5, and 0.7. We also train a fifth instance using a sample with all of those $r_{\mathrm{inv}}$ values mixed together. Each trained instance is then applied to samples with all four $r_{\mathrm{inv}}$ values, so each instance trained on a specific $r_{\mathrm{inv}}$ value must process signal events with different $r_{\mathrm{inv}}$ values. Figure 8 shows the results, with each network's output calibrated using the mixed $r_{\mathrm{inv}}$ sample for consistency. In general, we see that networks trained with lower values of $r_{\mathrm{inv}}$ also work well on signals with higher values of $r_{\mathrm{inv}}$. The networks trained with higher values of $r_{\mathrm{inv}}$ do not work quite as well on signals with lower values of $r_{\mathrm{inv}}$. This is expected, as some kinematic information is lost when $r_{\mathrm{inv}}$ increases; more dark hadrons are combined into the single $\vec{\not{p}}_{\mathrm{T}}$ two-vector rather than contributing to the jet four-vectors. The network trained on a mixed sample with all $r_{\mathrm{inv}}$ values works well for all $r_{\mathrm{inv}}$ values. This test demonstrates that the EVN can provide good mass reconstruction even when applied to signals with notably different parameter values than used in training, as long as the basic kinematic behavior is similar.

## 4.4 Out-of-band masses

A natural test of the EVN's generalization capabilities is exposing it to mediator mass values outside of the training dataset. To perform this test, we retrain the EVN on three subsets of the signal samples with different requirements on the $Z'$ mass: $m_{Z'} > 2500$; $m_{Z'} \leq 1500$ or $m_{Z'} > 3500$; and $m_{Z'} \leq 2500\,\mathrm{GeV}$. Because these restrictions significantly reduce the number of events available for training, the EVN layer sizes are reduced by a factor of 2 to 64, 32, 32, 32, 16, leading to a total of 6050 trainable parameters, and the batch size is reduced to 1000. Each trained network is calibrated using the full range of $m_{Z'}$ values for consistency. We observe in Fig. 9 that the EVN performs equally well on signals with mediator mass values below, in between, or above the range of mass values used for training, when compared to the EVN trained on the full range of available masses from Section 4.

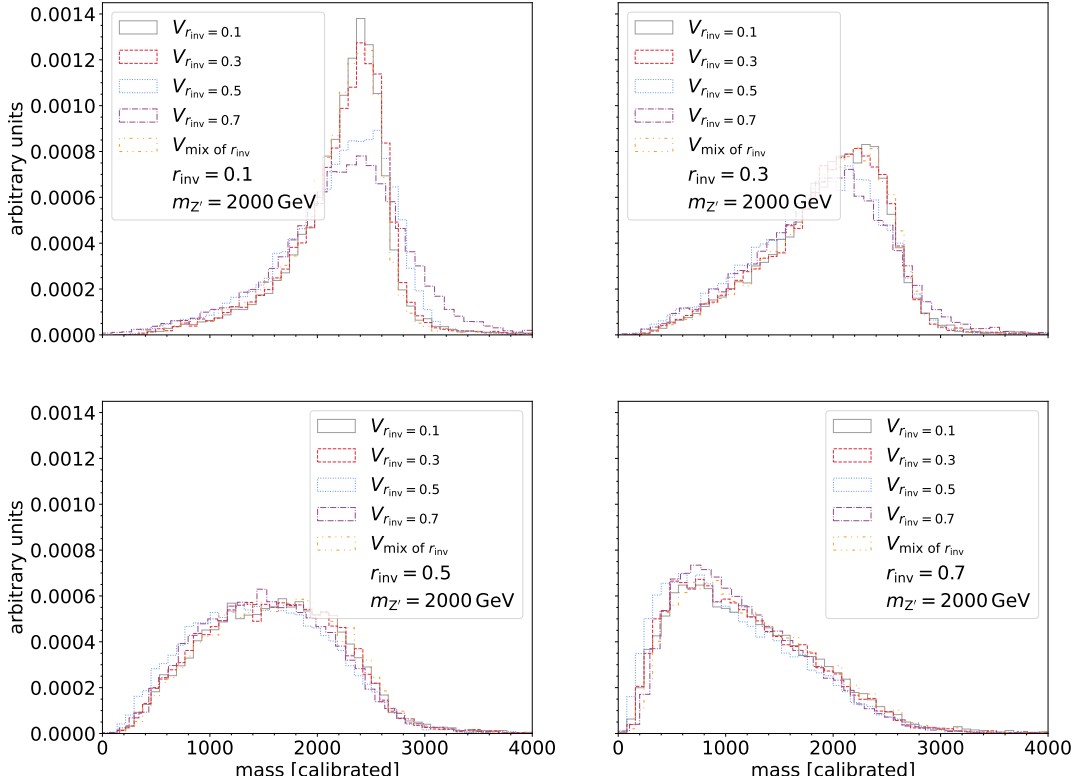

Figure 8: The distributions of the artificial variable $V$ from networks trained and tested on different $r_{\text{inv}}$ values. The subscripts in the $V$ entries in the legend indicate which $r_{\text{inv}}$ value was used for training, while the larger text at the bottom of each legend indicates which $r_{\text{inv}}$ sample was used to produce the distributions.

# 5 Φ production

## 5.1 Single production

The resonant production of a single bifundamental mediator Φ leads to a final state with two high-$p_{\text{T}}$ jets from the mediator decay—one SM jet and one semivisible jet—in association with a second low-$p_{\text{T}}$ semivisible jet. The transverse momentum distributions for the leading three jets in such events are shown in Fig. 10. Based on the kinematic differences between the two semivisible jets, the missing energy in the event is expected to be associated primarily with the higher-$p_{\text{T}}$ semivisible jet. Therefore, the classical approach to reconstruct the resonance mass is to compute the transverse mass from the two leading jets and the $\not{p}_{\text{T}}$, similar to the Z′ case. Because the other leading jet is a fully visible SM jet, there is no need to employ the MAOS procedure to split the $\not{p}_{\text{T}}$ into multiple components. However, it is possible that optimal usage of information from the third jet might improve the mass resolution; as a semivisible jet, it may contribute to the $\not{p}_{\text{T}}$.

Therefore, we train two versions of the EVN on single Φ production events. The first version uses just the two leading jet four-vectors and the $\not{p}_{\text{T}}$ two-vector as input, while the second version also includes the third jet four-vector. Because a smaller mass range is explored for Φ events, resulting in fewer training events, we use the alternative hyperparameter settings described in Section 4.4. The first network is compared to $m_{\Phi}$ and $M_{\text{T}}$ in Fig. 11, and the corresponding mass distributions are shown in Fig. 12 (left). It can be seen that the artificial variable does not exactly reproduce $M_{\text{T}}$ but highly correlates with it. Figure 12 (right) shows

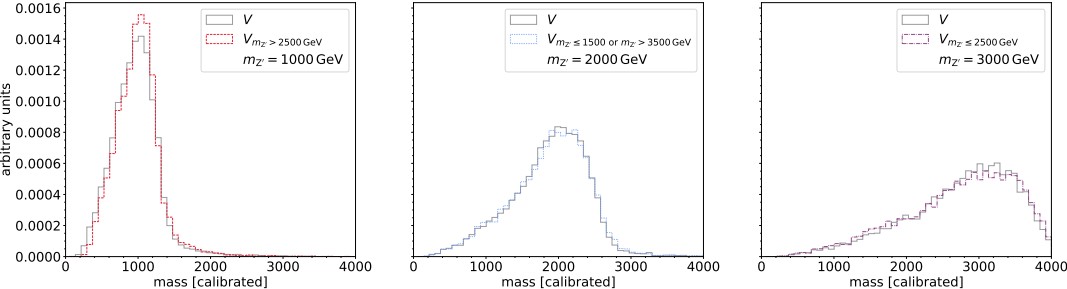

Figure 9: The distributions of the artificial variable $V$ from networks trained on different ranges of $m_{Z'}$ values. The subscripts in the $V$ entries in the legend indicate the $m_{Z'}$ range used for training (with no subscript used for the case where $V$ is trained on the entire range of masses), while the larger text at the bottom of each legend indicates which $m_{Z'}$ sample was used to produce the distributions.

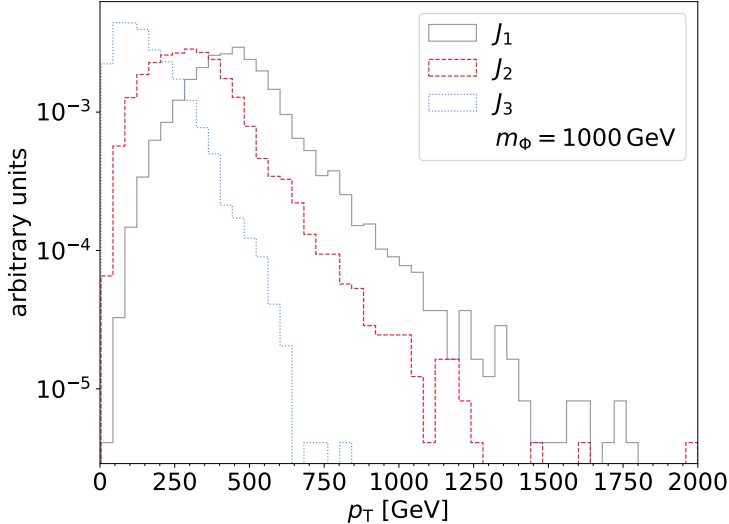

Figure 10: The $p_T$ distributions of the three leading jets in single production events with $m_\Phi = 1000$ GeV.

the correlation between the first and second versions of the network, which is similarly high. This indicates that including the third jet information does not improve the mass reconstruction for this topology. We conclude that the transverse mass is effectively optimal here.

## 5.2 Pair production

The pair production of $\Phi$ results in at least four jets, two of which are semivisible, along with missing energy. The classical approach to reconstruct the mediator mass in this final state is $M_{T2}$ or a similar variable. In this case, disambiguation must be performed, to attempt to combine the four jets into two visible four-vectors in a way that correctly pairs jets from the same mediator. Here, we use a standard method: choosing the pairing that minimizes the invariant mass difference between the two pairs, $\Delta M_{JJ} = |M_{J_a J_b} - M_{J_c J_d}|$. We also compare to the variable $M_{T2}^{\text{gen}}$, where the disambiguation is done correctly using generator-level information about the mediator decays, to represent an upper bound on possible performance.

The EVN is trained with the four-vectors of the leading four jets and the $\vec{\displaystyle{\not}p}_T$ two-vector as input. As in the single production case, the alternative hyperparameter settings from Section 4.4 are used. The results are shown in Figs. 13 and 14. The artificial variable is correlated with

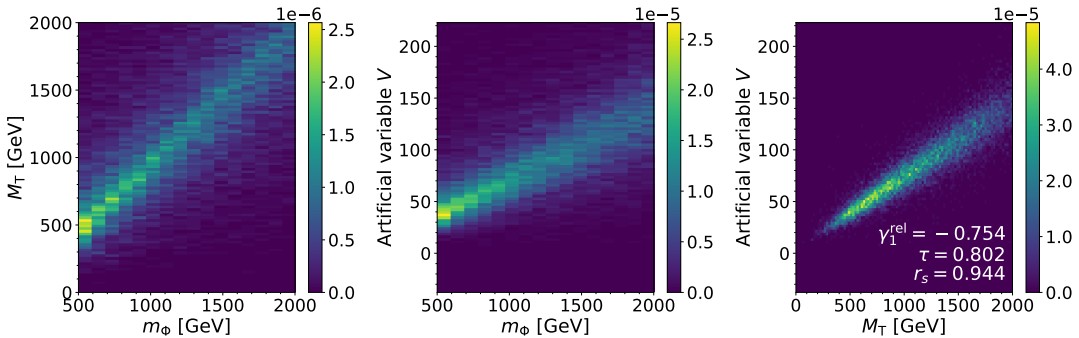

Figure 11: Correlations between $M_T$, $V$, and $m_\Phi$ in single production events.

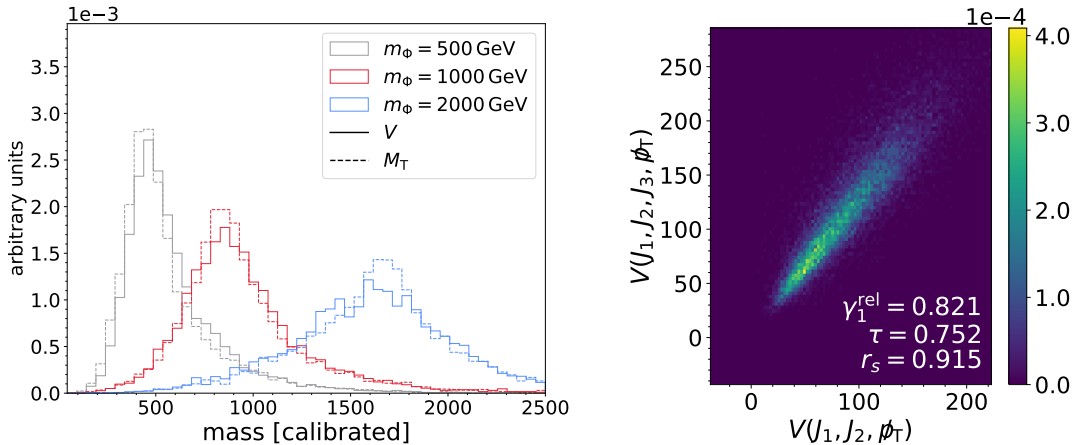

Figure 12: Left: the distributions of $M_T$ and $V$ for several $m_\Phi$ values in single production events, normalized to unit area. Right: the correlations between $V$ using two jets and $V$ using three jets.

$M_{T2}$, but is not identical to it. $M_{T2}$ has a substantial low-mass tail, which is typically caused by incorrect pairings from the $\Delta M_{JJ}$ heuristic. $V$, while it does not achieve the resolution of the generator-level version, largely eliminates this low-mass behavior.

To understand the impact of the apparent improvements of the artificial variable over the classical reconstruction, we conduct a sensitivity comparison, following the procedure and selection in Section 4.2. Figure 15 shows the result. We observe that, compared to $M_{T2}$, the artificial variable shifts the background to lower mass values and the signal to higher values, and both effects improve the sensitivity in the higher end of the mass distribution. The improvement is especially pronounced for higher $m_\Phi$ values, which are more challenging to discover in the LHC dataset because of their small cross sections. Computing the approximate overall significance shows that the artificial variable improves on $M_{T2}$ by 17% for $m_\Phi = 1000\,\text{GeV}$ and 81% for $m_\Phi = 2000\,\text{GeV}$. This again demonstrates the power of the semisupervised approach; the EVN, by learning an optimal generalized function for the signal topology, pushes the background away from resonant signal distributions without being trained on background kinematic distributions.

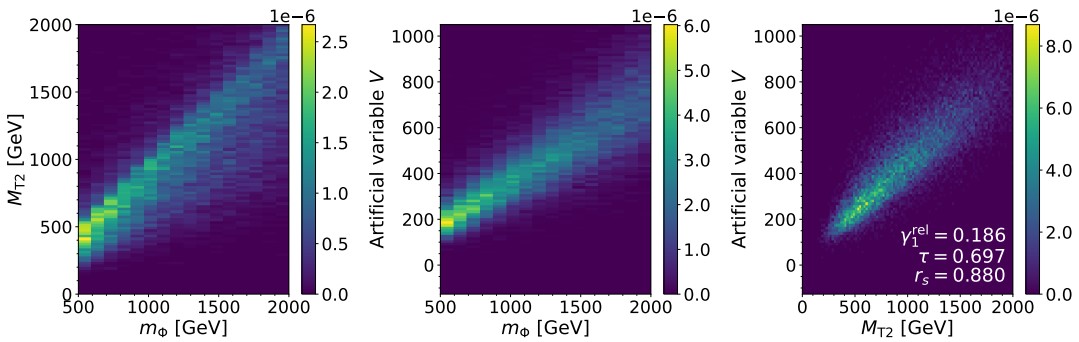

Figure 13: Correlations between $M_{T2}$, $V$, and $m_\Phi$ in pair production events.

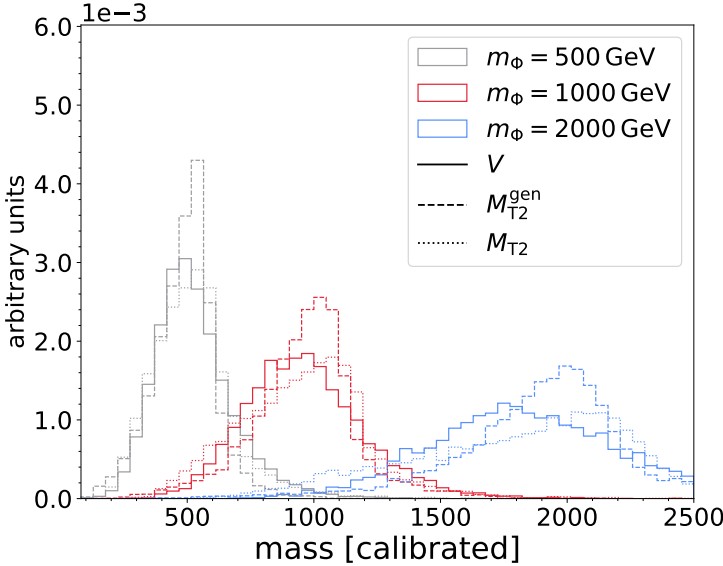

Figure 14: The distributions of $M_{T2}$, $M_{T2}^{gen}$, and $V$ for several $m_\Phi$ values in pair production events, normalized to unit area.

## 6 Conclusion

Semivisible jets are a novel phenomenological signature of strongly coupled dark matter, corresponding to stable dark hadrons that arise, along with unstable dark hadrons, from a hidden sector with a new, confining force. We apply the artificial event variable network (EVN) to achieve optimal mediator mass reconstruction for several resonant production channels: a Z′ boson, a single bifundamental scalar Φ, or a pair of Φ. This network uses an information bottleneck to produce interpretable output that is directly correlated with a given parameter, here the mediator mass. The EVN is trained with a semisupervised approach that uses only signal events, without any knowledge of standard model background processes.

In the Z′ case, we show that the learned artificial variable is superior to the transverse mass $M_T$ and similar to the $M_{T2}$-Assisted On Shell (MAOS) scheme. Using the artificial variable results in a moderate improvement, compared to the classical variables, in the sensitivity of a simplified search for this process. For this result, we compare signal events to quantum chromodynamics (QCD) multijet background events, for which the EVN produces a steeply falling mass distribution despite never being exposed to such events during training. We also use the Z′ case to further demonstrate the generalization capabilities of the EVN, showing that its performance generalizes to signal models with mediator mass values or invisible fractions

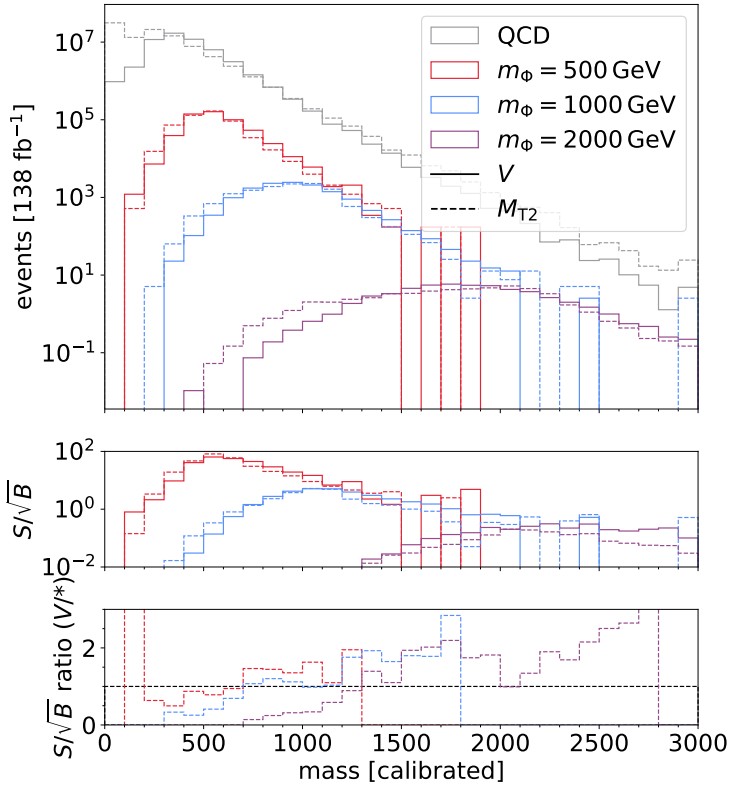

Figure 15: The distributions of each reconstructed mass variable for $\Phi$ pair production and QCD multijet samples. The middle pane shows the significance, and the bottom pane shows the ratio of the significance, comparing $M_{T2}$ to $V$. The distributions in the bottom pane are truncated to eliminate noisy values from statistical fluctuations in the high tails of the signal mass distributions.

(the proportion of dark hadrons that are stable) that differ from the training data.

We investigate the phenomenological behavior of single and pair production of $\Phi$ mediators for the first time. We show with the EVN that $M_T$ is approximately optimal for the single production case. In the pair production case, we find that the artificial variable significantly improves over the classical $M_{T2}$ and offers strong potential to improve the collider discovery reach for this process.

Overall, the artificial event variable network offers a promising avenue to improve event reconstruction in an interpretable and generalized way, by learning a function that correlates with physical information. It can be applied to improve sensitivity, increase discovery potential, and provide phenomenological knowledge even for complicated signals such as semivisible jet production.

## Acknowledgments

We thank Eshwen Bhal, Tim Cohen, Annapaola de Cosa, Sarah Eno, Aran Garcia-Bellido, Thomas Klijnsma, Kyoungchul Kong, Hou Keong Lou, Benedikt Maier, Sandeep Madireddy, Christopher Madrid, Konstantin Matchev, Siddharth Mishra-Sharma, Stephen Mrenna, Matt Strassler, and Chin Lung Tan for useful discussions. The color scheme used in the figures in this paper is intended to be accessible and was derived in Ref. [37].

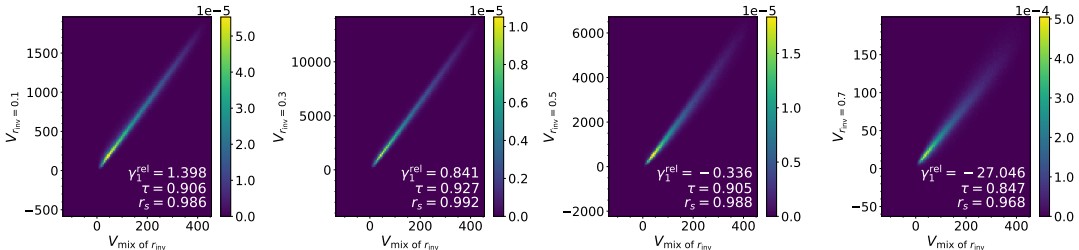

Figure 16: Correlations between $V$ trained on different $r_{\text{inv}}$ values and $V$ trained on a mix of $r_{\text{inv}}$ values in Z$'$ production.

**Funding information** K. Pedro and P. Shyamsundar are supported by Fermi Research Alliance, LLC under Contract No. DE-AC02-07CH11359 with the U.S. Department of Energy, Office of Science, Office of High Energy Physics. P. Shyamsundar is supported by the U.S. Department of Energy, Office of Science, Office of High Energy Physics QuantISED program under the grants "HEP Machine Learning and Optimization Go Quantum", Award Number 0000240323, and "DOE QuantiSED Consortium QCCFP-QMLQCF", Award Number DE-SC0019219.

# A  Additional information

## A.1  Code availability

The code used to train the neural network and produce the figures in this paper can be found at https://github.com/kpedro88/evn_svj_public. The MADGRAPH and PYTHIA settings used to generate the signal and background events can also be found there.

## A.2  Supporting distributions

### A.2.1  Z$'$ production

Figure 16 shows correlations between artificial variables trained using signals with different $r_{\text{inv}}$ values. High correlations with the artificial variable trained on all $r_{\text{inv}}$ values are observed. The relative skewness $\gamma_1^{\text{rel}}$ shows that variables trained on low $r_{\text{inv}}$ have moderate biases toward high values compared to the mixed training case and variables trained on high $r_{\text{inv}}$ have moderate biases toward low values, as expected.

Figure 17 shows correlations between artificial variables trained using different ranges of $m_{\text{Z}'}$ values. High correlations with the artificial variable trained on the full range of $m_{\text{Z}'}$ values are observed. The relative skewness $\gamma_1^{\text{rel}}$ shows that the variable trained on low $m_{\text{Z}'}$ has moderate bias toward low values compared to the full range case and the variable trained on high $m_{\text{Z}'}$ has moderate bias toward high values, as expected.

### A.2.2  $\Phi$ production

Figure 18 shows the distribution of $M_{\text{T}}$ and the artificial variable trained using three jet four-vectors and the $\vec{\not{p}}_{\text{T}}$ two-vector, rather than the default two jet four-vectors and $\vec{\not{p}}_{\text{T}}$, in single $\Phi$ production events. The distribution is very similar to the variable using the default inputs in Fig. 12.

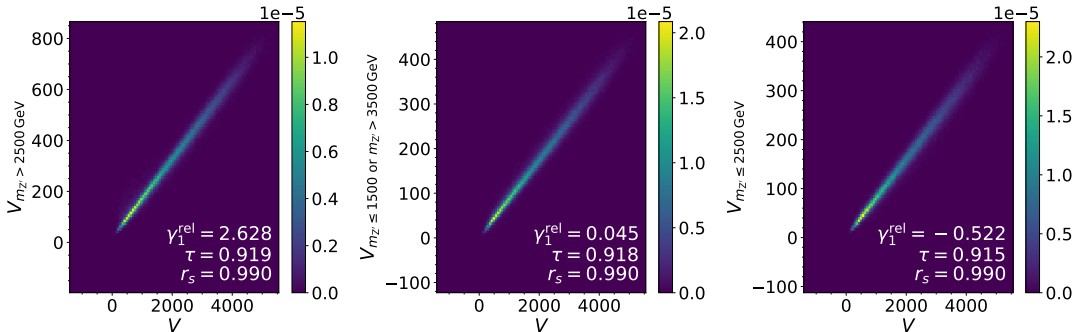

Figure 17: Correlations between $V$ trained on different ranges of $m_{Z'}$ values and $V$ trained on a the full range of $m_{Z'}$ values.

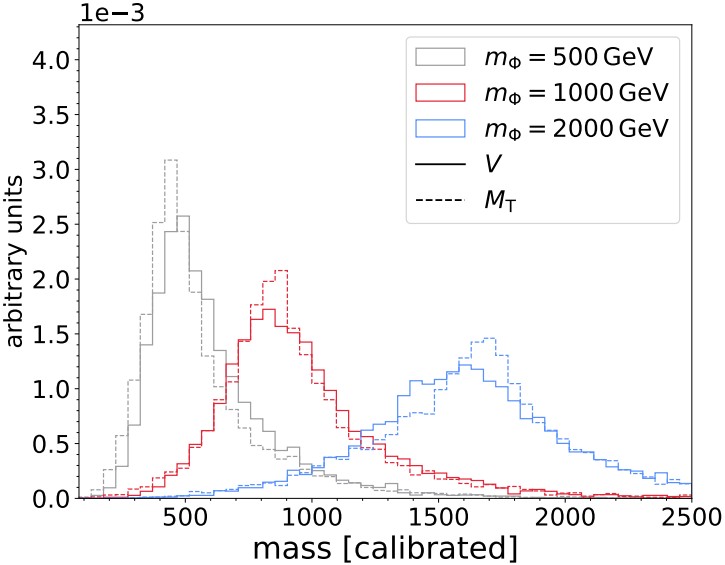

Figure 18: The distributions of $M_T$ and $V$ trained with three jets for several $m_\Phi$ values in single production events, normalized to unit area.

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
