# Peer review of "Optimal Mass Variables for Semivisible Jets"

_SciPost Physics Core, doi:SciPost Phys. Core 6, 067 (2023)_

## Round 2 · Referee Report · Anonymous · 2023-7-6

Strengths
1. This paper studies the interesting question of improving collider searches for semi-visible jets. This is a motivated model with distinct collider signatures for which dedicated and novel searches can significantly improve the potential for discovery.
2. The paper presents a novel approach using ML tools to searching for semi-visible jets and conclusively shows that this approach is superior to the current state of the art.
3. This paper explores the applicability of this method to several signal models and demonstrates that their method is applicable and effective for all of them.
4. This paper explores the possibility of searching for signals outside the parameter space of the training data, and shows that their method works very well in that case.
5. The paper is clearly and logically written.
Report
As noted above, the paper concerns an important and interesting subject, and quantitatively shows that their proposed method is an improvement over the state of the art. I believe, however, that some revisions are required before the paper can be published.
1. The main results plots for two of the production models, figs 6, 15, show the improvement relative to state of the art methods in the third panel. These distributions, particularly in fig 6, show some values below 1 and some values above 1. I request that the authors estimate how much improvement this method will give in reach and/or sensitivity of a search to a particular model.
2. In figure 4, why is the fit done to a line with no offset? Namely, why do they use y=mx instead y = mx+b. Also in figure 4, the middle column seems to have a positive correlation between the variables, so why is m < 0?
3. The authors say that the R_T cut is important for reducing background. As such, the authors should explain in more detail how their cut differs from the experimental papers they cite, and how exactly they expect their different cut can compensate for a lack of detector simulation.
4. The authors should explain in detail what chi is in the Feynman diagrams of fig 1. I believe it is meant to be a dark quark which then showers and hadronizes into a jet of dark hadrons, but that should be in the text.
5. There is a mistake in definition of sensitivity at the beginning of section 4.2, there is a slash missing.
6. I'm not sure I understand the point of figure 2. The authors are not looking at signals with different numbers of phi's in the same signal region. Furthermore, the ratio of cross section depends very sensitively on the Yukawa couplings, so I'm not sure this plot is very meaningful for one value.
Author: Kevin Pedro on 2023-07-18 [id 3819]
(in reply to Report 1 on 2023-07-06)
We thank the referee for their thoughtful review of our paper. Our responses to the revisions requested by the referee can be found below. The updated version of the manuscript is available at https://arxiv.org/abs/2303.16253v3 and has been resubmitted to SciPost Phys.
-
We computed an estimate of the overall significance for each case and added the following descriptions to the paper: At the end of Section 4.2: “The overall significance for each signal model and mass variable can be approximated by adding the significance of each bin in quadrature. This calculation shows that the artificial variable has a 3--5% higher significance than MT, while MAOS has only a ~1% higher significance.” At the end of Section 5.2: “Computing the approximate overall significance shows that the artificial variable improves on MT2 by 17% for mΦ = 1000 GeV and 81% for mΦ = 2000 GeV.”
-
We find that the offset b in the y=mx+b fit is very small and can be neglected. The negative sign was a mistake, which has been fixed. (In some cases, the artificial variable can have a negative correlation with the theory variable. In these cases, we multiply the artificial variable by -1, which is why all the plots show a positive correlation. We had forgotten to remove the minus signs from the values of m displayed in the plots.)
-
We added more detail to the explanation of the RT cut: “Therefore, we apply a selection RT > 0.1, which is looser than the experimental thresholds of 0.15--0.25. The lower ptmiss values in the generator-level background imply that fewer events have high RT values, so this looser selection provides a level of background rejection similar to the realistic analysis.”
-
We added the definition of the symbol χ where dark quarks are first mentioned in the text (Section 1, paragraph 2) and elaborated in the caption of Fig. 1: “Representative Feynman diagrams for leading-order production of a Z′ boson decaying to dark quarks χ (left), a single Φ boson associated with a dark quark and decaying to an SM quark and a dark quark (center), and a pair of Φ bosons each decaying to a dark quark and an SM quark (right).”
-
Fixed.
-
We included Fig. 2 to put each Φ production mode in context. While the optimal signal regions for each production mode are different, their absolute and relative cross sections illustrate their expected abundance in collider production. A realistic search strategy would likely consider all possible production modes, so it is useful to understand how many events would be expected in each case. In addition, the collider phenomenology of these different production modes was not thoroughly studied until this paper, and so it is our view that it is useful to introduce some basic properties of these signal models. In particular, these plots complement and extend Fig. 10 (left) from 1707.05326. While it is true that the cross sections are dependent on the Yukawa couplings, our choice of coupling value is clearly stated in Section 2.
Author: Kevin Pedro on 2023-07-18 [id 3820]
(in reply to Report 2 on 2023-07-12)We thank the referee for their thoughtful review of our paper. Our responses to the revisions requested by the referee can be found below. The updated version of the manuscript is available at https://arxiv.org/abs/2303.16253v3 and has been resubmitted to SciPost Phys.
Other uses of ML for SVJs and studies of SVJ characteristics pertain to jet tagging and substructure. Those topics are not relevant to this work, which focuses on event-level mass reconstruction. We did not cite either our own or others’ work on such topics.
We elaborated on this point in Section 1, paragraph 6: “In particular, Ref.~\cite{Kim:2021pcz} shows that for a fully visible final state where the target parameter is the theoretical mediator mass, the learned function is equivalent to the invariant mass calculation, and therefore we expect the EVN to produce optimal estimators in semivisible final states, as well.”
Resonant searches typically are not binned in MET or jet pT; indeed, as high-mass resonance signatures include multiple jets, the latter quantity may be ambiguous. A basic discussion of the performance of conventional mass variables such as MT can be found in the literature. The high correlation between the learned variable and the conventional variables (shown in e.g. Fig. 4 for Z′) entails that the different variables must have similar performance. Therefore, basic physical principles will be followed in all cases, such as better resolution when jets have higher pT.
Fixed.
Our results show that even training on a mixed sample is unnecessary; the semisupervised nature of the training procedure automatically eliminates dependence on rinv (to the extent possible). Therefore, further investigation of more complicated decorrelation techniques was also deemed to be unnecessary.
Changed to “QCD multijet background”, “standard model hadrons”, and “may include”.
While the referee’s point about model dependence is correct in general, this study is actually less sensitive than most to effects from dark sector parameter variations. Those variations largely influence the showering and hadronization in the dark sector, with potential impact on the flavors and kinematic behavior of SM hadrons in the final state. However, this study does not consider jet substructure or other such variables, only high-level jet four-vectors and the ptmiss two-vector. The mediator mass and rinv are the only parameters that significantly affect those quantities, and the variations of those parameters are considered in great detail. A remark was already made about this in Section 2, paragraph 1: “The dark hadron mass scale does not significantly impact the event-level kinematic quantities that are used in this paper, so we do not consider variations of it or the parameters with related values.”
We added some clarification about this point to Section 2, paragraph 2: “The HV module simulates the dark sector dynamics using the Lund string model~\cite{Andersson:1983ia,Sjostrand:1984ic}; we use the default settings for the empirical parameters in this model, which are the values tuned for SM QCD. Different parameter values, or even a different dynamical model or generator software, could change the dark sector dynamics. However, such variations generally impact the formation and substructure of jets, rather than their final four-momenta, and therefore are not expected to have a significant impact on the event-level mass reconstruction pursued here.”

---

## Round 2 · Referee Report · Anonymous · 2023-7-12

Strengths
1. Exploration of strongly interacting dark sector has received a lot of attention lately, however the experimental searches are often limited by similarity of the signal with dominant multijet background. So any step towards improving discrimination is a step in the right direction.
2. Rather than restricting themselves to the most commonly studied Z' mediator, authors also considered two other production modes.
3. The paper did show possible gain in sensitivity.
4. I would also commend the authors for putting the code on github for scrutiny and use by the community.
Weaknesses
1. This is not so much a weakness of this paper per se, but as it stands, this study is somewhat Pythia8 HV model, and specific parameter choice dependent.
2. It is not obvious a priori that what EVN is outputting is mass.
3. It is slightly concerning that only 6000 or 12000 events per signal point were generated/used.
4. The network/result depends possibly on rinv.
5. Its not clear, if there is any dependence on MET or jet pT.
Report
Firstly I would like to apologise for the delay. June/July is a busy month with meetings and holidays.
This is a useful study, and as such I think deserves to be published. Some details can be clarified a bit more to improve the quality of the paper. Some suggestions are below.
Requested changes
1. The paper seems to be missing citing some other recent SVJ work, especially the ATLAS result and other studies about characteristics of SVJ and use of ML methods.
2. It would be appreciated if the authors explain a bit more how EVN output is the mass, not just a mass-like observable.
3. Did the authors look at the performance as a function of MET or the jet pT? It will be good to add that.
4. In Sec 4.2, the significance is given as Q = S times sqrt(B), surely a division is missed?
5. Did the authors think of making the procedure less dependent on rinv just apart from training on a mixed sample?
6. Minor text changes: QCD background -> multijet background, regular hadrons (in abstract) reads weird, final state will include SVJ (3rd paragraph in Sec 1)-> may include.
7. Also it will be useful to include a statement on the dependence on Pythia HV and lack of other HV modules.

---

## Round 3 · Referee Report · Anonymous (Referee 1) · 2023-7-19

Report

I would like to thank the authors for their careful consideration of my previous report. I am satisfied with their responses to points 1, 3, 4, and 5. I would like to request further changes regarding points 2 and 6. Please see below.

Requested changes

Regarding point 2 in my previous report, the authors should explain in the text why there is no offset in the fit of figure 4, as they did in their response.

Regarding point 6, I disagree that figure 2 puts the different searches in context given that the rates are strongly dependant on the Yukawa coupling, and the value that they have chosen is no more motivated than any other value. If the authors do want to keep figure 2, they should explain how the curves scale with the Yukawa couplings, and qualitatively describe what the figure would look like for a few other values of the coupling.

  • validity: -
  • significance: -
  • originality: -
  • clarity: -
  • formatting: -
  • grammar: -

Author:  Kevin Pedro  on 2023-07-21  [id 3827]

(in reply to Report 1 on 2023-07-19)

2\. We added a note about this after the next-to-last sentence in the second paragraph of Section 4.1: “(The constant term in the linear fit is found to be very small and is therefore neglected.)”

6\. We added the requested context where the figure is mentioned in paragraph 3 of Section 2: “...shown in Fig. 2 for the chosen Yukawa coupling value y_dark = 1.0. Nonresonant production depends more strongly on y_dark than single production, while pair production primarily depends only on α_S; therefore, the relative fractions of the resonant production modes would increase for smaller y_dark values and decrease for larger y_dark values.”

---

## Round 3 · Author Response

This resubmission addresses feedback from the two referees.

---

## Round 3 · List of Changes

• abstract: "regular hadrons" -> "standard model hadrons"
  • Section 1, paragraph 2: defined dark quark symbol χ where dark quarks are first introduced
  • Section 3, paragraph 3: "will include" -> "may include"
  • Figure 1, caption: now reads "Representative Feynman diagrams for leading-order production of a Z′ boson decaying to dark quarks χ (left), a single Φ boson associated with a dark quark and decaying to an SM quark and a dark quark (center), and a pair of Φ bosons each decaying to a dark quark and an SM quark (right)."
  • Section 1, paragraph 5: added citation of ATLAS result, "A recent search from the ATLAS experiment sets limits on certain models of non-resonant $t$-channel SVJ production via Φ~\cite{ATLAS:2023swa}, but does not target resonant production."
  • Section 1, paragraph 6: added sentence "In particular, Ref.~\cite{Kim:2021pcz} shows that for a fully visible final state where the target parameter is the theoretical mediator mass, the learned function is equivalent to the invariant mass calculation, and therefore we expect the EVN to produce optimal mass estimators in semivisible final states, as well."
  • Section 2, paragraph 2: added sentences "The HV module simulates the dark sector dynamics using the Lund string model~\cite{Andersson:1983ia,Sjostrand:1984ic}; we use the default settings for the empirical parameters in this model, which are the values tuned for SM QCD. Different parameter values, or even a different dynamical model or generator software, could change the dark sector dynamics. However, such variations generally impact the formation and substructure of jets, rather than their final four-momenta, and therefore are not expected to have a significant impact on the event-level mass reconstruction pursued here."
  • Figure 4: fixed errant negative sign in fit parameter value in legend
  • Section 4.2, paragraph 1: fixed typo in sensitivity definition, now "S/√B".
  • Section 4.2, paragraph 1: added sentence "The lower ptmiss values in the generator-level background imply that fewer events have high RT values, so this looser selection provides a level of background rejection similar to the realistic analysis."
  • Section 4.2, paragraph 1: changed "QCD background" to "QCD multijet background".
  • Section 4.2, paragraph 1: added sentences "The overall significance for each signal model and mass variable can be approximated by adding the significance of each bin in quadrature. This calculation shows that the artificial variable has a 3--5\% higher significance than \MT, while \MAOS has only a ${\approx}1\%$ higher significance." and changed "improvement" to "improvements" in the following sentence.
  • Section 5.2, paragraph 3: added sentence "Computing the approximate overall significance shows that the artificial variable improves on MT2 by 17% for mΦ = 1000 GeV and 81% for mΦ = 2000 GeV."

---

## Round 4 · Author Response

This resubmission addresses followup comments from one of the referees.

---

## Round 4 · List of Changes

• Section 1, paragraph 3: fix typo "will may" -> "may include"
  • Section 2, paragraph 3: add information about Yukawa coupling dependence: “...shown in Fig. 2 for the chosen Yukawa coupling value y_dark = 1.0. Nonresonant production depends more strongly on y_dark than single production, while pair production primarily depends only on α_S; therefore, the relative fractions of the resonant production modes would increase for smaller y_dark values and decrease for larger y_dark values.”
  • Section 4.1, paragraph 2: add a note about the calibration fit: “(The constant term in the linear fit is found to be very small and is therefore neglected.)”

---

## Editorial Decision

published